# Fosfomycin Resistance Evolutionary Pathways of *Stenotrophomonas maltophilia* in Different Growing Conditions

**DOI:** 10.3390/ijms23031132

**Published:** 2022-01-20

**Authors:** Teresa Gil-Gil, José L. Martínez

**Affiliations:** 1Programa de Doctorado en Biociencias Moleculares, Universidad Autónoma de Madrid, 28049 Madrid, Spain; tgil@cnb.csic.es; 2Centro Nacional de Biotecnología, CSIC, Darwin 3, 28049 Madrid, Spain

**Keywords:** *Stenotrophomonas maltophilia*, experimental evolution, fosfomycin resistance

## Abstract

The rise of multidrug-resistant Gram-negative pathogens and the lack of novel antibiotics to address this problem has led to the rescue of old antibiotics without a relevant use, such as fosfomycin. *Stenotrophomonas maltophilia* is a Gram-negative, non-fermenter opportunistic pathogen that presents a characteristic low susceptibility to several antibiotics of common use. Previous work has shown that while the so-far described mechanisms of fosfomycin resistance in most bacteria consist of the inactivation of the target or the transporters of this antibiotic, as well as the production of antibiotic-inactivating enzymes, these mechanisms are not selected in *S. maltophilia* fosfomycin-resistant mutants. In this microorganism, fosfomycin resistance is caused by the inactivation of enzymes belonging to its central carbon metabolism, hence linking metabolism with antibiotic resistance. Consequently, it is relevant to determine how different growing conditions, including urine and synthetic sputum medium that resemble infection, could impact the evolutionary pathways towards fosfomycin resistance in *S. maltophilia*. Our results show that *S. maltophilia* is able to acquire high-level fosfomycin resistance under all tested conditions. However, although some of the genetic changes leading to resistance are common, there are specific mutations that are selected under each of the tested conditions. These results indicate that the pathways of *S. maltophilia* evolution can vary depending on the infection point and provide information for understanding in more detail the routes of fosfomycin resistance evolution in *S. maltophilia*.

## 1. Introduction

Antibiotic resistance is nowadays one of the main problems for public health [1], and regarding this threat, the dissemination of multidrug-resistant Gram-negative bacteria is of great concern. One of these Gram-negative pathogens is *Stenotrophomonas maltophilia*, an opportunistic nosocomial pathogen causing different infections and characterized by its low intrinsic susceptibility to broad-range antibiotics [2]. Infections associated with *S. maltophilia* include serious nosocomial infections such as respiratory tract infections, bacteremia, biliary sepsis, bones, joints, soft tissues, or eye infections, endophthalmitis, dacryocystitis, endocarditis, meningitis and urinary tract infections (UTIs) [3,4]. However, UTIs caused by this microorganism have been rarely reported [5]. Although *S. maltophilia* can acquire antibiotic resistance genes through horizontal gene transfer (HGT) [6], mutation is the main cause for the emergence of antibiotic resistance by this organism, mainly when causing chronic infections in cystic fibrosis patients [7,8,9].

The increased resistance of Gram-negative bacteria to currently used antibiotics, together with the lack of new antibiotics, has renewed the interest in old less-used antibiotics such as fosfomycin [2], which is being used as a first-line treatment for the treatment of UTIs and has been proposed for treating other infections [10,11]. Fosfomycin binds to MurA (UDP-N-acetylglucosamine enolpyruvyl transferase), which catalyzes the first step in peptidoglycan biosynthesis and inactivates it, leading to bacterial cell lysis [12]. Although *S. maltophilia* is considered intrinsically resistant to fosfomycin, previous studies have shown that current methods for measuring fosfomycin susceptibility, which include glucose-6-phosphate in their formulation, are not accurate enough for this microorganism (Gil-Gil and Martinez, submitted). Further, it has been shown that the phenotype of antibiotic resistance may depend on the environment—and hence on the infection point [13]—as well as on the conditions of selection [14]. Consequently, it is of relevance to search the mechanisms of fosfomycin resistance that *S. maltophilia* can acquire under different conditions, including urine and synthetic cystic fibrosis sputum medium (SCFM), which are representative of infective conditions.

It was previously shown that the main causes of acquiring fosfomycin resistance in different bacteria are mutations in the genes encoding the fosfomycin target MurA [15] or its transporters, UhpT (glucose-6-phosphate transporter) and GlpT (glycerol-3-phosphate transporter) [16,17], as well as the activity of fosfomycin-inactivating enzymes such as the glutathione transferase FosA [18]. However, none of these mechanisms have been previously reported to be involved in the acquisition of increasing levels of fosfomycin resistance in single-step *S. maltophilia* mutants. In this microorganism, mutations in genes encoding enzymes of the central metabolism are on the basis of fosfomycin acquired resistance; in particular, mutations that inactivate glycolytic enzymes belonging to the Embden–Meyerhof–Parnas (EMP) metabolic pathway [19] are responsible for the acquisition of resistance. This indicates that *S. maltophilia* fosfomycin resistance is linked to the bacterial metabolism, a feature also supported by recent data showing that changes in the transcriptome of *S. maltophilia* when challenged by either fosfomycin, phosphoenolpyruvate (PEP) or glyceraldehyde-3-phosphate are very similar [20]. Given the observed close relationship between fosfomycin resistance and *S. maltophilia* metabolism, we challenged *S. maltophilia* on experimental evolution studies with increasing concentrations of fosfomycin in different media—LB, urine and SCFM—and analyzed the mutations arising after each evolution. These evolutions allowed the determination of genetic changes underlying fosfomycin antibiotic resistance in different conditions, including laboratory growing conditions and situations that mimic those that *S. maltophilia* find when producing urinary or lung infections. Our results reinforce the concept that metabolic changes can alter the patterns and phenotypes of evolution towards antibiotic resistance [21].

## 2. Results

### 2.1. Experimental Evolution upon Fosfomycin Challenge Leads to High Levels of Resistance in S. maltophilia Populations under all Tested Conditions

In order to elucidate how growing conditions may impact the acquisition of fosfomycin resistance by *S. maltophilia*, four independent *S. maltophilia* D457 populations were serially passaged for 72 h in the presence of increasing concentrations of fosfomycin in LB, urine or SCFM until a concentration of 4 minimum inhibitory concentration (MIC) was reached. In parallel, four independent control replicates were also passaged under the same conditions in the absence of antibiotics. It is worth mentioning that *S. maltophilia* growth increased urine pH from 7 to 8.

Afterwards, MICs for the fosfomycin-evolved populations, as well as the wild-type strain D457 and the control-evolved populations, were determined by double dilution in MH, LB, urine and SCFM. As shown in Table 1, all populations exhibited high-level resistance to fosfomycin in comparison with the parental strain D457 and the control-evolved populations, reaching at least MIC increases of 16-fold compared with the parental strain. In addition, fosfomycin-evolved populations show a fosfomycin resistance phenotype in all tested media, both in those where they have evolved and in those where they did not. These results suggest that the acquired mutations are able to cause resistance in all tested media, no matter in which medium they have been selected.

### 2.2. Selected Mutations in the Presence of Fosfomycin

Whole-genomic DNA of evolved populations in the presence of fosfomycin and of the controls, grown in the absence of the antibiotic, as well as the DNA of the parental strain were extracted and sequenced, with the aim of determining the genetic changes potentially responsible for the decreased susceptibility to fosfomycin. Only those mutations that were selected upon antibiotic selective pressure but were absent in the populations evolved in the absence of selection were taken into consideration and are listed in Table 2. In order to confirm the presence of the genetic changes identified by whole-genome sequencing (WGS), the sequences containing the different mutations were amplified and Sanger-sequenced from stored samples of each population. Altogether, genetic modifications were found in different genes, single-nucleotide polymorphisms (SNPs) causing amino acid substitutions being the most common changes.

The four populations that evolved in LB in the presence of fosfomycin shared a deleterious mutation in *eno*, which encodes a phosphopyruvate hydratase (Eno), with a prevalence of 93–98% within the analyzed populations. The finding of this SNP in the *eno* gene reinforces previous results showing that the mutational inactivation of different enzymes belonging to the EMP central metabolism pathway, including Eno, is on the basis of fosfomycin resistance acquisition in *S. maltophilia* [19]. In addition, in population B a genetic change was observed in *phaR*, encoding a polyhydroxyalkanoate synthesis repressor, even though the percentage of reads that contain the variation within the heterogeneous population is only 22%.

Moving to urine-evolved populations, all of them share two genetic changes. Firstly, the four of them present the same mutation in *eno* that has been previously found in the LB evolution. Nevertheless, the frequency of this mutation drops to 9–21% in the case of urine-evolved populations; this fact could explain the presence of extra mutations found in this medium unlike LB to reach the same level of resistance. Secondly, a one-base insertion in *virB10* (SMD_RS13785) was detected in the four populations. VirB10 is one of the proteins forming the type IV secretion system (T4SS) outer membrane channel, which traverses both the inner and outer membrane, playing an important role in the regulation of substrate transfer to the extracellular space [22]. Mutations in this gene have been previously linked to enhanced susceptibility to vancomycin [23]. This may suggest that a possible change in the outer membrane permeability due to this mutation allows bidirectional movement. substrate release to cell surface and uptake into the periplasm, of molecules too large to diffuse through porins across the outer membrane. To ascertain if this mutation alters the activity of vancomycin as described, the susceptibility of the fosfomycin-evolved populations to this antibiotic was determined. No changes were observed with respect to the parental strain or the evolved strains in the absence of antibiotic, all of them presenting a vancomycin MIC of 256 μg/mL. Last, an SNP in SMD_RS05475 encoding a BolA family transcriptional regulator in populations A and C was also detected. Even though this mutation does not produce an amino acid change and is considered neutral, its genomic location, in the same operon and upstream *murA*, the only known fosfomycin target, might suggest that this SNP could affect *murA* expression (see below).

Genetic modifications in five different genes were found in SCFM-evolved populations, all of them sharing the same mutation in *bolA* previously found in urine-evolved populations and an 18 bp insertion that results in a genomic variant upstream of *recR*. In addition to these common changes, population A and B show an SNP in *gspL*, which encodes a PilN domain-containing protein which takes part in the type II secretion system (T2SS). T2SS of *S. maltophilia* promotes bacterial virulence and hence human infection, but has not been previously related to antibiotic resistance [24]. Aside from previously detailed changes, population A presents two extra SNPs. The first of them is located in SMD_RS14605, encoding a protein that is part of an ABC transporter that shares a 97.83% identity with lipid A export ATP-binding/permease inner membrane protein MsbA of *Pseudomonas aeruginosa*. This protein is involved in lipid A transport across the cytoplasmic membrane from the inner to the outer leaflet of the inner membrane [25,26,27,28,29]; its deficiency has been associated with hypersensitivity to glutaraldehyde, erythromycin and hydrophobic antibiotics, such as rifampicin, in *Helicobacter pylori* and *Lactobacillus lactis* [30,31]. The last change in population A was found in *ptsN*, a nitrogen regulatory protein paralogous to the enzyme IIA^fru^ of the PEP-dependent carbohydrate phosphotransferase system (PTS) [32,33]. PtsN is also known as Enzyme IIA^Ntr^, taking part in a phosphorelay system (PTS^Ntr^) in *Escherichia coli* that may have a role in the regulation of nitrogen metabolism [34,35] and that regulates the activity and expression of potassium transporters [36,37,38].

### 2.3. Antibiotic Susceptibility Profile of the Fosfomycin-Evolved Populations

To explore the influence of mutations selected under fosfomycin pressure on cross-resistance and/or collateral sensitivity to other antibiotics, MICs of antibiotics belonging to different classes were measured at the end of each evolution (Figure 1).

In all analyzed conditions, some fosfomycin-evolved populations with a decreased susceptibility to colistin—LB-evolved populations A, B and C; urine B and D; and SCFM A and B—and polymyxin B—urine-evolved populations C and D—were found. Further, low susceptibility to ceftazidime was observed in the urine-evolved C population. The increased resistance to colistin observed in most populations may be associated with a change in permeability, as this antibiotic interacts electrostatically with the outer membrane of Gram-negative bacteria, displaces divalent cations which stabilize the lipopolysaccharide (LPS) layer and thus disrupts the membrane integrity and diffuses through it inside the cell [39].

Regarding collated sensitivity, susceptibility to some antibiotics was detected. Population LB A presented collateral susceptibility to amikacin, and urine A and C to trimethoprim/sulfamethoxazole (SXT), which is of relevance since this antibiotics combination is the first-line treatment for *S. maltophilia* infections. Similar increased SXT susceptibility is observed, even if it is not of significant value, in all fosfomycin-evolved populations, suggesting that the administration of fosfomycin together with SXT for the treatment of *S. maltophilia* infections might be taken into consideration.

### 2.4. Fitness Costs Associated with Fosfomycin Resistance

The relative fitness costs to the 12 fosfomycin-evolved populations were assessed by measuring their growth rates and comparing them with that of the D457 wild-type strain in each medium. Firstly, in LB laboratory medium, LB-evolved populations showed around 11 to 20% deficiency in their growth rate, 17 to 44% in urine populations and around 5 to 7% deficiency in SCFM evolutions (Figure 2A). On the one hand, these results indicate that LB-evolved populations present a better growth rate than those that have evolved in urine, although SCFM populations present a better growth rate, indicating that populations presenting mutations in genes encoding enzymes of the central metabolism have a greater relative fitness cost. On the other hand, *virB10* mutations could be the cause of the differences in fitness cost between urine and LB-evolved populations. Secondly, when tested in urine, LB-evolved populations had a growth deficiency from 10 to 14% and SCFM-evolved populations present values ranging from 6 to 13%. However, urine-evolved populations were not defective in their growth. Notably, they presented a greater growth rate than the D457 wild-type strain (Figure 2B); suggesting that selected mutations in this medium, apart from increasing resistance to fosfomycin, allow a better growth in the selection medium. It is then possible that the one-pair insertion in *virB10* gene would be selected for improving the growth of *S. maltophilia* in urine. Thirdly, the growth in SCFM of all the evolved populations shows a deficiency with respect to the parental strain, reaching from 52 to 65% in the populations evolved in LB, 14 to 39% in the case of those evolved in urine—whose differences do not seem to be related to the selected mutations in each of them—and between 22 and 34% in those evolved in SCFM (Figure 2C). Taken together, these results show that the selection of mutations in each evolution is occasionally—as in the case of urine—but not always related to a growth improvement in the medium in which the experimental evolution takes place.

### 2.5. Expression Changes in Possible Resistance Determinants in the Fosfomycin-Evolved Populations

Apart from the mutation found in the *eno* gene described as a fosfomycin resistance mechanism in our bacterium of interest, most of the genetic changes detected by WGS did not show a direct correlation with fosfomycin resistance mechanisms previously described for other bacterial species. To ascertain if these mutations produce changes in the expression level of these genes, their expression was analyzed by quantitative reverse-transcription PCR (qRT-PCR) (Figure 3). As mentioned, *bolA* is transcribed in an operon with *murA* gene. It has been previously assessed that an increased expression of *bolA* occurs under cell envelope stress conditions and its deletion produce changes in the outer membrane, making it more permeable and increasing susceptibility to vancomycin and fosfomycin [40,41]. However, a significant change in the expression of *bolA* was not observed; neither in the populations that present the mutation in that gene nor in any other population (Figure 3A). Despite the lack of changes in the expression of this gene, and since mutations were located upstream of the gene encoding the fosfomycin target *murA*, the expression of this gene was measured. Our results show that fosfomycin-evolved populations with a mutation in *bolA* gene exhibit greater *murA* expression (Figure 3B). Hence, the enhanced expression of this gene may be the main reason for the fosfomycin resistance of these fosfomycin-evolved populations, since an increased synthesis of this enzyme confers a resistant phenotype [42]. To further confirm this issue, MurA activity was also measured in all evolved populations. Consistent with the results dealing with *murA* expression, we observed that those populations with a mutation in *bolA* presented increased MurA activity compared with the wild-type strain (Figure 4). These results confirm that *murA* overexpression is on the basis of the resistance phenotype of these populations.

Regarding *virB10*, urine A and D fosfomycin-evolved populations showed an increased expression of this gene (Figure 3C). VirB10-defective mutants have been previously proved to exhibit enhanced sensitivity to molecules that do not normally cross the outer membrane through porins, such as vancomycin [23]. Notwithstanding, as previously mentioned, these populations do not present changes in the susceptibility to this antibiotic. All the same, since the fosfomycin gates of *S. maltophilia* are not known, it is still possible that mutations and/or changes in the expression of the T4SS would be related to fosfomycin resistance, besides its role in virulence.

A common WGS change found in all SCFM-evolved populations is an 18 pb insertion located upstream of *recR*, encoding a protein involved in DNA recombination. In order to determine if this change produces a direct effect on *recR*, the expression of this gene was measured. However, no changes were observed in its level of expression (Figure 3D). Finally, no changes were observed in the levels of expression of *gspL*, SMD_RS14605 or *ptsN* in any of the SCFM-evolved populations, including the SCFM linages A and B in which mutations in these genes were found (Figure 3E–G).

### 2.6. Activity of Central Metabolism Enzymes Is Associated with Fosfomycin Resistance in Some of the Fosfomycin-Evolved Populations

The only mechanism of acquired resistance to fosfomycin described to date in *S. maltophilia* is the impaired activity of enzymes of the lower glycolysis pathway [19]. Our results suggest that some of the fosfomycin-evolved populations could share this resistance mechanism. To determine whether the lower glycolytic pathway is involved in the fosfomycin resistance of the fosfomycin-evolved populations, the enzymatic activity of glyceraldehyde-3-phosphate dehydrogenase (Gap), phosphoglycerate kinase (Pgk), phosphoglycerate mutase (Gpm), Eno and pyruvate kinase (Pyk) was measured in the evolved-populations and in the wild-type strain. Moreover, the activity of the main dehydrogenase glucose-6-phosphate dehydrogenase (Zwf) was measured to determine the general physiological state of the populations.

As shown in Figure 5, in LB-evolved populations, Gap, Eno and Pyk activities decrease in all four populations in relation to the parental strain (Figure 5A,D,E). Thus, Eno mutation causes a loss of function of the encoded protein and a metabolic rearrangement that modifies the mutated pathway. Urine-evolved populations share the same *eno* mutation with LB-evolved populations, although no changes were observed in the activity of this enzyme (Figure 5D). The low proportion of this mutation in the study populations could be the cause of the lack of change in Eno activity. To ascertain if this was the reason for the observed results, individual clones from each urine-evolved population were isolated on LB agar plates and those presenting the *eno* mutation were chosen for further analysis. The selected clones had a resistance value to fosfomycin identical to that of the population they were selected from. As shown in Figure 6, isolated clones presented defects in *eno* activity in comparison with the wild-type strain. This last result shows that the reason why a deficiency in the activity of *eno* was not found in the urine-evolved populations was indeed the low proportion of this mutation in such populations. In addition, Pgk activity decreased in urine C and D fosfomycin-evolved populations (Figure 5B). In addition, Zwf activity, which is a key element in the bacterial oxidative response [43], increased with a significant change in all urine-evolved populations compared with the wild-type D457 strain (Figure 5F). These results suggest that urine-evolved populations show a different physiological state than the wild-type strain.

Finally, the resistance mechanisms of SCFM-evolved populations do not seem to be related to a metabolic shift, showing the same general physiological state as the parental D457 strain (Figure 5).

All in all, both single-step *S. maltophilia* fosfomycin-resistant mutants [19] and fosfomycin-evolved populations, in both cases obtained in LB medium, present similar mechanisms of resistance to fosfomycin, the inactivation of enzymes belonging to the central carbon metabolism, while the resistance mechanism in clinical settings, urine and SCFM, appears to be not solely related to a metabolic shift. In urine, the resistance mechanism appears to be multifactorial and related to changes in central metabolism activities, whereas in the most relevant environment in infections caused by *S. maltophilia*, SCFM, the cause of resistance is not related, at least in the first instance, to changes in the central metabolism.

### 2.7. Fosfomycin Resistance Is not Related to an Impaired Entry of Fosfomycin Inside the Cell

Despite the fact that the *S. maltophilia* genome does not harbor genes encoding either the canonical fosfomycin transporters or already known fosfomycin-inactivating enzymes, it might be possible that other elements may contribute to impairing the accumulation of the antibiotic inside the studied populations. Additionally, the only common cross-resistance change among most populations is the resistance to colistin, an antibiotic that diffuses through membrane. It may be also possible that mutations in the secretion system, observed in urine and SCFM-evolved populations, produce changes in the permeability of the outer membrane.

To analyze the possibility of an impaired antibiotic transport, the intracellular accumulation of fosfomycin in the different populations was measured. In any fosfomycin-evolved populations, the amount of intracellular fosfomycin was lower (Figure 7). These results confirm that the fosfomycin resistance is not due to a reduced intracellular concentration of this antibiotic in any of the fosfomycin-evolved populations. Further, an increased intracellular fosfomycin concentration was observed in some populations, being statistically significant in SCFM A- and B-evolved populations. These populations share an SNP in the gene *gspL*, which is not mutated in any other population. This gene encodes a fimbrial assembly protein, an inner membrane component of the T2SS that provides a link between the energy-providing GspE protein in the cytoplasm and the rest of the T2SS machinery [24]. This mutation in the secretion systems could change outer membrane permeability leading to an enhanced uptake of molecules across the outer membrane into the periplasm. In addition, the SNP observed in SMD_RS14605 in lineage SCFM A could be also related to an increased intracellular fosfomycin concentration. Despite this increase in the intracellular concentration of fosfomycin, both populations show resistance to this antibiotic, which suggests that other resistance mechanisms, such as the higher concentration and activity of MurA (see above), are capable of producing resistance despite the increase in the intracellular concentration of fosfomycin.

## 3. Discussion

Although it has been described that fosfomycin resistance is associated with a high biological cost in the absence of fosfomycin, which may lead to the low prevalence of fosfomycin-resistant strains [44], using in vitro experimental evolution we have determined the first adaptive events driven by the exposure of *S. maltophilia* to increasing concentrations of fosfomycin in different settings, both laboratory and clinical. The evolutionary changes leading to fosfomycin resistance have been shown to be different in each condition and similar in the four parallel cultures of each medium. We have proved that even though evolutionary trajectories are different depending on the medium in which bacteria are subjected to a selective pressure, all selected mutations are able to cause resistance in all tested media. Moreover, as far as our knowledge reaches, none of the mutations have been associated with fosfomycin resistance in other bacterial pathogens besides *S. maltophilia*.

Fosfomycin resistance mechanisms described so far include the three classical categories—alterations in fosfomycin transport, antibiotic inactivation and alterations in the target enzyme or peptidoglycan biosynthesis [17,45]—along with the recently described inactivation of enzymes belonging to the EMP pathway. This last category was described after the study of one-step laboratory-isolated mutants obtained on LB agar medium with a supra-inhibitory fosfomycin concentration [19]. Our results show that experimental evolutions in LB medium result in mutations in one of the previously described enzymes, the inactivation of which leads to fosfomycin resistance: the enzyme enolase from the lower-glycolytic metabolic pathway. This mutation impairs the enolase activity, together with a decrease in the activity of the glyceraldehyde-3-phosphate dehydrogenase and the pyruvate kinase lower-glycolysis enzymes. All these facts strongly suggest that mutations of genes encoding metabolic enzymes are the cause of *S. maltophilia* acquiring fosfomycin resistance in laboratory media.

Our study enables us to find different evolutionary trajectories depending on the medium in which the selection is carried out; due to the fact that no possible fosfomycin transporter has been found in the *S. maltophilia* genome, it allows finding alternative routes to the mutation of the canonical transporters of this antibiotic as the first acquired mechanism of resistance [46]. WGS analysis has indicated that mutations in genes encoding lower-glycolysis enzymes are also on the basis of the acquisition of resistance in urine. Despite the fact that *eno* mutations are found in a small proportion inside the populations without changing enolase activity at the population level, urine-evolved populations show a different physiological state as evidenced by an increased Zwf activity, an enzyme that supplies the cofactor NADPH needed to maintain cellular redox balance [47,48]. Together with this mutation, all urine-evolved populations present a one-pair insertion in the *virB10*-encoding gene. Even though mutations in this gene have been previously related to an increase in antibiotic susceptibility [23], and the present mutation seems to be deleterious for the gene, the results strongly suggest that mutations in T4SS are related to fosfomycin resistance in urine, since this mutation is present in all four linages. Furthermore, *bolA* mutation may be related to an increased synthesis and activity of the only known fosfomycin target, the enzyme MurA. Additionally, BolA homologs have been previously shown to be involved in acid and cell envelope stressors resistance at the same time that their deletions lead to reduced peptidoglycan and altered outer membrane lipids, leading to fosfomycin and vancomycin susceptibility [40,41]. Regardless, the mutation found is not deleterious to BolA, which leads us to think that its participation in *S. maltophilia* fosfomycin resistance is related to an increase in the expression of MurA; a fact that should be studied in more detail.

Previous analyses have shown that fosfomycin activity is higher in urine under acidic conditions [49]. In acidic urine (pH 5 to 6), fosfomycin is partially protonated and enters bacteria, resulting in higher fosfomycin activity. Therefore, urine acidification increases fosfomycin susceptibility even in the case of resistance isolates, including high-level fosfomycin-resistant mutant strains. At pH 5 most strains are susceptible to fosfomycin. In UTIs caused by *E. coli*, urine is mainly acidic, with pH levels lower than 6.5, a fact that could explain the low prevalence of fosfomycin-resistant *E. coli* in these infections [49]. We detected an increased alkalization in urine from a pH value of 6–7 to 8 after 24 h of *S. maltophilia* growth. This alkaline pH might compromise fosfomycin activity and would allow the selection of mutations conferring low-level fosfomycin resistance. Despite this possibility, our results illustrate that urine-evolved populations are fosfomycin resistant in all tested media apart from urine. It is clear that *S. maltophilia*-selected mutations in urine award high-level fosfomycin resistance. However, it is also worth stating that, as above mentioned, UTIs caused by this microorganism have been rarely reported, a feature that somehow might reduce the clinical impact of our findings.

Moving to the most relevant clinical environment where *S. maltophilia* infections may occur, cystic fibrosis, first acquired fosfomycin resistance in this SCFM is definitely not related to a general metabolic shift. All four lineages present two common mutations: an SNP in *bolA* that has also been found in urine and an insertion upstream of *recR*. Since four SCFM and two urine lineages present a *bolA* mutation, it is clear that this not-deleterious mutation is the basis of fosfomycin-acquired resistance in clinically relevant media. In addition, these populations show an increased synthesis and expression of MurA, leading to fosfomycin resistance.

Apart from these common mutations, SCFM lineages A and B present extra mutations that do not significantly increase the level of fosfomycin resistance compared with the other two lineages. Firstly, a single-nucleotide variant in *gspL* was found in both A and B lineages. This protein plays a role in the complex assembly and recruits GspM, resulting in a stable complex in the inner membrane [24]. Mutations in secretion system genes could lead to permeability changes in the outer membrane, as has been previously mentioned, a fact that explains why these two populations present a higher intracellular fosfomycin concentration than their parental strain. This kind of mutation enhances sensitivity to molecules that normally inhibit cell wall synthesis and do not enter the cell through porins, being consistent with this last result. Despite this fact, these two populations present a fosfomycin resistance phenotype and do not show collateral sensitivity to antibiotics such as vancomycin. Secondly, population A presents a SNP in *SMD_RS14605* gene and another in *ptsN* gene, both related to transport across the membrane. In addition, *SMD_RS14605* is homologous to the *msbA* gene of *P. aeruginosa*, which has been previously related to antibiotic susceptibility.

MsbA has been suggested to be involved in active efflux, as well as in LPS production, carrying out the first essential step in the trafficking of LPS to the outer membrane [50]. Thus, its deficiency decreases LPS production, increasing outer membrane permeability, and consequently enhances membrane permeability for the toxic substrates moving in, including hydrophobic antibiotics that are not pumped out by MsbA [31]. All these facts suggest that a deleterious mutation of the gene encoding this ABC inner transport protein would increase susceptibility to antibiotics such as aminoglycosides or chloramphenicol [51,52,53,54,55]; a fact that we did not observe. In addition, its deficiency would increase the intracellular concentration of these antibiotics it pumps out, suggesting that fosfomycin would be one of these antibiotics even though the increased intracellular fosfomycin concentration does not result in an associated increased susceptibility.

The acquisition of all these mutations may imply changes in susceptibility to other antibiotics apart from fosfomycin. To test this possibility, the susceptibility to other antibiotics belonging to different families was assessed in all the fosfomycin-evolved populations. Not many changes were observed, suggesting that the selected mutations specifically confer resistance to fosfomycin. However, two important changes attract attention: cross-resistance to colistin, which could suggest a membrane deficiency and collateral susceptibility to SXT, the most used antibiotic for treating *S. maltophilia* infections [6]. This last change suggests that the use of fosfomycin in combination with SXT could be a useful alternative for treating SXT-resistant *S. maltophilia* infections. These results show that the different media result in similar outcomes regarding susceptibility to other antibiotics not used in selection.

One important factor influencing the emergence of antibiotic-resistant bacteria is the resistance-associated fitness cost. Even though the acquisition of resistance implies a fitness advantage in the presence of the antibiotic of selection, in the absence of the selection, pressure-resistant bacteria would be outcompeted by susceptible ones [56,57]. Our data show that in the absence of the antibiotic of selection, all evolved populations present a fitness cost in LB and SCFM media. However, when tested in urine, even though LB- and SCFM-evolved populations displayed a fitness cost, urine-evolved populations presented a greater growth than the wild-type strain. All in all, in the absence of antibiotics it could be difficult for most fosfomycin-resistant mutants to be maintained in the population [58].

Our results once again show the importance of the interlink between the modification of the activity of enzymes belonging to central metabolism and the antibiotic susceptibility, which should be added as a new molecular mechanism of fosfomycin resistance, together with those traditionally described. In addition, we have been able to find fosfomycin resistance adaptive pathways that do not include permeability changes as the initial step due to a mutation in the genes encoding canonical fosfomycin transporters. In the same way, our results prove that the acquisition of resistance is different depending on the medium used, obtaining different main resistance mechanisms depending on the medium in which bacteria are subjected to the antibiotic pressure. Our results then highlight the need to use conditions that mimic actual bacterial infections in the search for resistance mutational trajectories by experimental evolution.

## 4. Materials and Methods

### 4.1. Growth Conditions and Antibiotic Susceptibility Assays

Bacteria were grown using liquid LB (lysogeny broth) Lennox medium, sterile urine from 6 healthy people (3 men and 3 women) that were between 25 and 55 years old and had not received antibiotics in the year before, or SCFM [19] at 37 °C with constant agitation at 250 rpm. Evolution experiments were performed with *S. maltophilia* D457, a model strain that had been isolated from a bronchial aspirate [59]. Fosfomycin (Sigma) was used at different concentrations during the evolution period. Antibiotic susceptibility testing was carried out in both the wild-type strain *S. maltophilia* D457 and all the fosfomycin-evolved populations. The MICs of fosfomycin and vancomycin were determined by double dilution in 96-well microtiter plates (NunclonTM Delta Surface) in LB at 37 °C. MICs of amikacin, streptomycin, tetracycline, tigecycline, ciprofloxacin, ofloxacin, aztreonam, nalidixic acid, gentamicin, ceftazidime, chloramphenicol, trimethoprim/sulfamethoxazole, colistin and polymyxin B were determined using MIC test strips (Liofilchem) on Mueller–Hinton (MH) agar plates at 37 °C for 20 h. Overnight bacterial cultures were normalized to an OD_600_ of 2, and a 1:1000 dilution of each culture was inoculated in the test plates.

### 4.2. Experimental Evolutions

Experimental evolutions were performed with the wild-type D457 strain growing in LB medium, urine or SCFM in the presence of increasing concentrations of fosfomycin. Eight independent experimental evolutions in each medium (four in the presence of fosfomycin and four controls without antibiotic) were carried out over a period of 3 days. Fosfomycin-challenged *S. maltophilia* D457 populations were initially grown at the maximum concentration that allowed bacterial growth in these conditions (128 μg/mL in LB, 256 μg/mL in urine and 512 μg/mL in SCFM). Five hundred microliters of a bacterial overnight culture were inoculated in flasks containing 20 mL of the corresponding medium. Serial passages were performed every 24 h inoculating 500 μL of bacterial cell cultures in fresh medium containing twice the concentration of fosfomycin, until fosfomycin concentration increased 4-fold. Controls were also cultured in the same conditions but in the absence of antibiotic. Samples were preserved at −80 °C every 24 h.

### 4.3. DNA Extraction, WGS and Identification of Mutations

Total genomic DNA from the evolved populations and the wild-type D457 strain were extracted using a Gnome^®^ DNA kit following the manufacturer’s protocol (MP Biomedicals). Quality and quantity of the extracted DNA was determined by agarose gel electrophoresis and using a NanoDrop Spectrophotometer, respectively. WGS was performed by Macrogen Inc. using the HiSeq 2000 Sequencing System (Illumina) generating 150 bp paired-end reads. The number of reads per sample was 5,000,000 on average, which represents a sequencing coverage of approximately 100×. Illumina short reads (2 × 150 bp) were checked with FASTQC (Simon Andrews’s group. https://www.bioinformatics.babraham.ac.uk/projects/fastqc/ Last access date: 15 December 2021). The *S. maltophilia* D457 reference genome (NC_017671.1) was used to align the WGS obtained using RNA STAR [60] (—alignIntronMax = 1). SNPs and small insertions/deletions (INDEL) were detected using freebayes (--strict-vcf --pooled-continuous-P 1). Only primary alignments, not marked as duplicated, were considered (‘effective reads’). Impact of detected SNP and INDEL was evaluated using snpEff [61]. Genomic regions with no coverage (large deletions) were detected using bedtools genomecov command [62]. The given variants were filtered against those obtained for the wild-type D457 strain and visualized by SNPer (https://bioinfogp.cnb.csic.es/tools/snper_dev/ Last access date: 15 December 2021). The presence of each putative mutation was verified by polymerase chain reaction (PCR) using the corresponding pairs of primers (Appendix A). DNA fragments were purified using the QIAquick PCR Purification kit (QIAGEN, Hilden, Germany) and Sanger sequencing was performed at GATC Biotech.

### 4.4. Fitness Cost Determination

Bacterial samples from the fosfomycin-evolved populations were used for the assay. A 10 μL sample of each culture was inoculated in 140 μL of LB medium, urine or SCFM to a final OD_600_ of 0.01 using 96-well plates (Nunc MicroWell; Thermo Fisher; Waltham, MA, USA). Growth (OD_600_) from three independent replicates was monitored every 10 min using a Spark 10 M plate reader (Tecan; Männedorf, Switzerland) for 24 h at 37 °C. Shaking for 10 s was performed before each measurement. OD_600_ values at the exponential growth phase were used to calculate the area under the curve. Relative growth rates were calculated by dividing the values of area under the curve of each population by those obtained for the parental strain D457 from the same experiment.

### 4.5. RNA Extraction and Quantitative Reverse Transcription PCR (qRT-PCR)

*S. maltophilia* D457 wild-type and all the evolved populations were grown overnight in LB broth at 37 °C and 250 rpm. These cultures were used to inoculate new flasks to reach an optical density at 600 nm (OD_600_) of 0.01, and the cultures were grown at 37 °C until an OD_600_ of 0.6 was reached. RNA was isolated following the protocol described by Gil-Gil T et al. [19]. DNA contamination was checked by PCR with primers 27 and 48 (Appendix A). Only RNAs containing no DNA contamination were used for further studies. Using a high-capacity cDNA reverse transcription kit (Applied Biosystems, Waltham, MA, USA), cDNA was obtained from 10 μg of RNA. qRT-PCR was carried out with a Power SYBR green PCR master mix (Applied Biosystems) in an ABI Prism 7500 real-time system (Applied Biosystems). A total of 50 ng of cDNA was used in each reaction, except for the wells with no template used as negative controls. A first denaturation step at 95 °C for 10 min was followed by 40 cycles at 95 °C for 15 s and 60 °C for 1 min for amplification and quantification. Primers that amplify specific fragments of the desired genes were designed with Primer3 Input software (v. 0.4.0) and were used at 400 nM (Appendix A). The primers gyrA_F and gyrA_R were used to amplify the housekeeping gene *gyrA* (Appendix A). Differences in the relative amounts of mRNA were determined according to the 2^−ΔΔCT^ method [63]. In all cases, the values of relative mRNA expression were determined as the average of three independent biological replicates, each containing two technical replicates.

### 4.6. Assay of MurA Activity

Cells were grown until exponential phase (OD_600_ = 0.6) and harvested by centrifugation at 5100× *g* and 4 °C and washed twice in 0.9% NaCl and 10 mM MgSO4. Once washed, cells were resuspended in 1 mL of 100 mM Tris [pH 7.5]–5 mM MgCl2–1 mM dithiothreitol (DTT), disrupted by sonication at 4 °C, and the cell extracts were obtained by centrifugation at 23,100× *g* for 30 min at 4 °C. Protein concentration was determined following the Pierce BCA Protein assay kit (Thermo Scientific; Waltham, MA, USA) protocol in 96-well plates (Nunc MicroWell; Thermo Fisher; Waltham, MA, USA).

The assay mixture (final volume, 72 μL) contained 50 mM Tris at pH 7.5, 2 mM DTT and 10 mM UDP-GlcNAc, and 200 μg/mL of bacterial extract protein. Samples were preincubated for 15 min at 37 °C, and the reaction was started by the addition of 8 μL of 10 mM PEP. After 2 h of incubation at 37 °C, 20 μL of color reagent (bicinchoninic acid solution and Copper (II) sulfate solution) was added to stop the reaction and measure the release of orthophosphate (Pi), monitored every 30 min using a Spark 10 M plate reader (Tecan) for 2 h. Results are expressed as OD_620_ values corrected for the background reading in the absence of UDP-GlcNAc.

### 4.7. In Vitro Activity Assays of the Enzymes of the Lower Glycolytic Pathway and Dehydrogenases

Protein extracts and protein quantification were performed as above mentioned. NAD(P)+ reduction or NAD(P)H oxidation was monitored spectrophotometrically at 340 nm and 25 °C with intermittent shaking in microtiter plates using a Spark 10 M plate reader (Tecan; Männedorf, Switzerland). Each reaction was performed using three biological replicates, and the specific activities were obtained by dividing the measured slope of NAD(P)H formation or consumption by the total protein concentration. Enzymatic activities of dehydrogenases (Zwf and Gap) were measured as described previously [19]. Enzymatic activities of Pgk, Pgm, Eno and Pyk were assayed following the protocol described by Gil-Gil, T et al. [19] in a two-step reaction.

### 4.8. Quantification of Intracellular Fosfomycin

Assays to test fosfomycin accumulation in bacterial cells were conducted as previously stated [19]. For measuring the amount of intracellular concentration of fosfomycin, fosfomycin-evolved populations, *S. maltophilia* D457 wild-type strain and *P. aeruginosa* PA14 wild-type strain, as well as *P. aeruginosa* Δ*glpT* and Δ*fosA* as controls, were used. The fosfomycin concentration is represented as micrograms per 10^7^ cells.

## Figures and Tables

**Figure 1 ijms-23-01132-f001:**
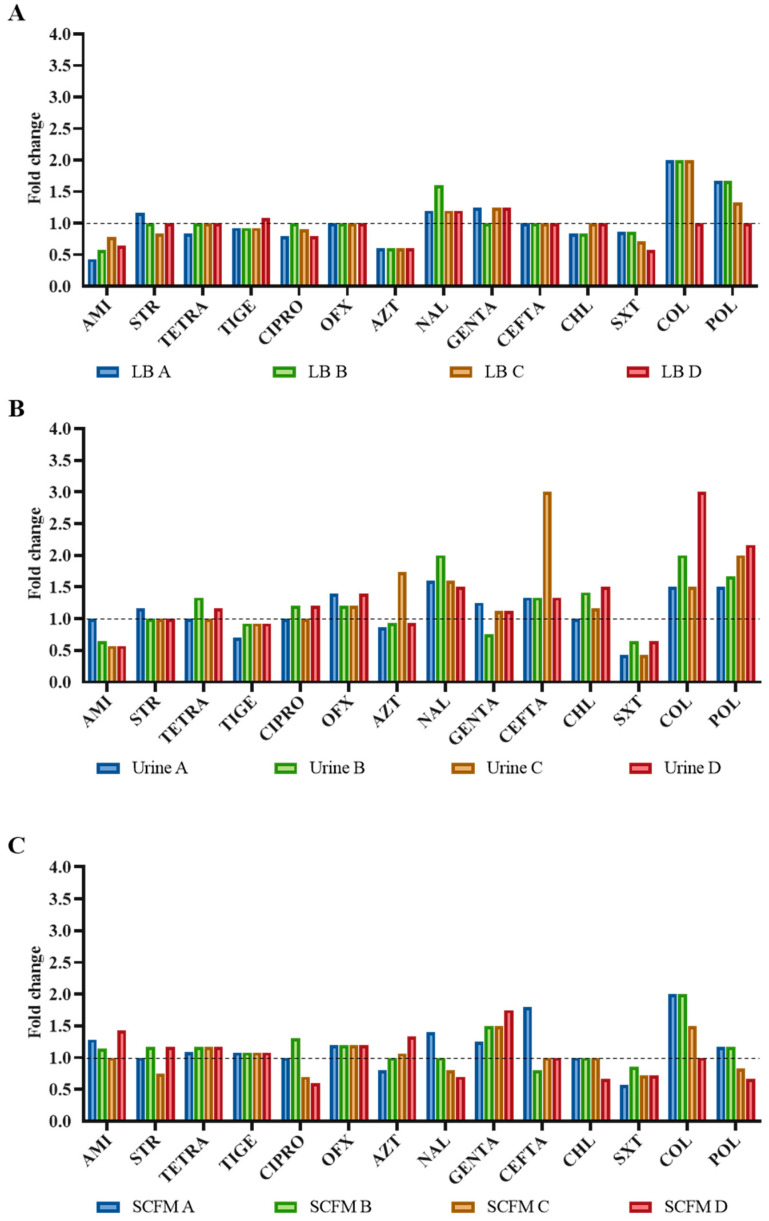
Collateral susceptibility of the *S. maltophilia* evolved populations to antibiotics from different families. MICs were determined in the twelve fosfomycin-evolved populations using the MIC values of the wild-type strain D457 as reference (dashed line). (**A**) LB-evolved populations; (**B**) urine-evolved populations; (**C**) SCFM-evolved populations. Values of at least double or half the wild-type MIC were considered significant. AMI, amikacin; STR, streptomycin; TETRA, tetracycline; TIGE, tigecycline; CIPRO, ciprofloxacin; OFX, ofloxacin; AZT, aztreonam; NAL, nalidixic acid; GENTA, gentamicin; CEFTA, ceftazidime; CHL, chloramphenicol; SXT, trimethoprim/sulfamethoxazole; COL, colistin; POL, polymyxin B.

**Figure 2 ijms-23-01132-f002:**
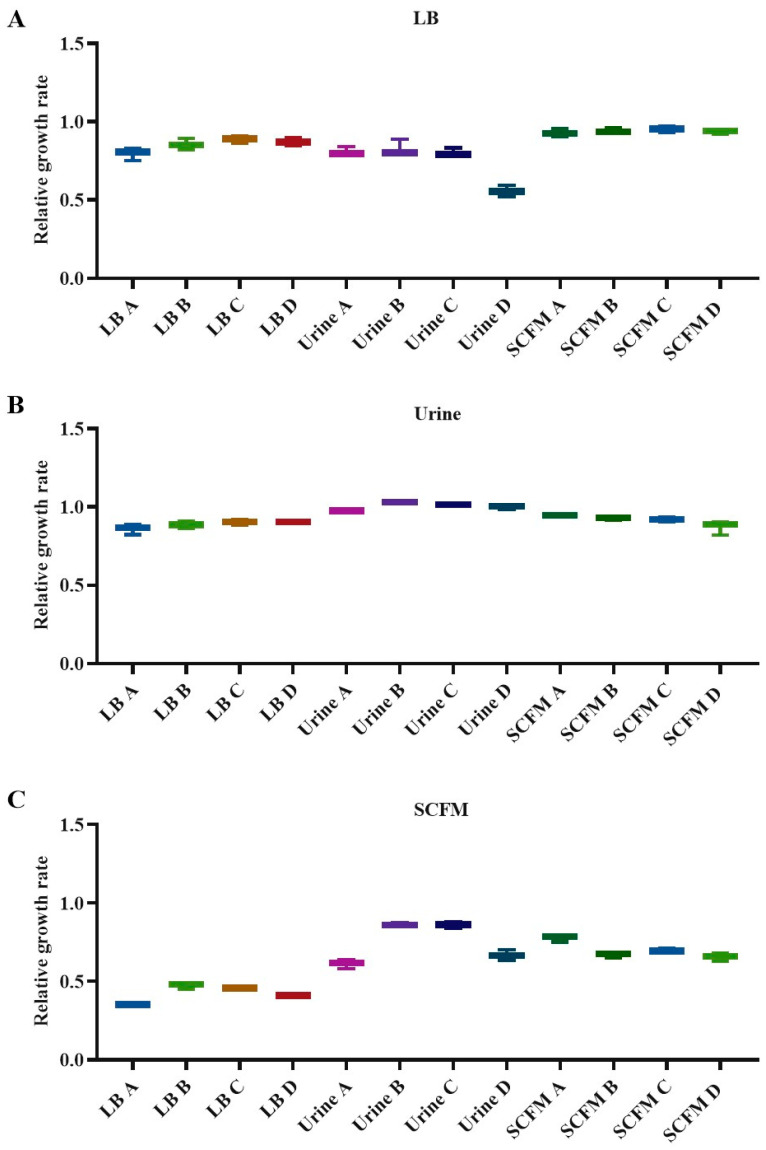
Fitness costs of the fosfomycin-evolved *S. maltophilia* populations. Growth experiments were performed in the four evolved populations. Growth rates were calculated from OD_600_ values corresponding to exponential growth. Relative growth rates were calculated using the wild-type D457 value as a reference. Growth was measured in LB (**A**), urine (**B**) and SCFM (**C**) mediums. Error bars represent the standard deviation from three independent replicates.

**Figure 3 ijms-23-01132-f003:**
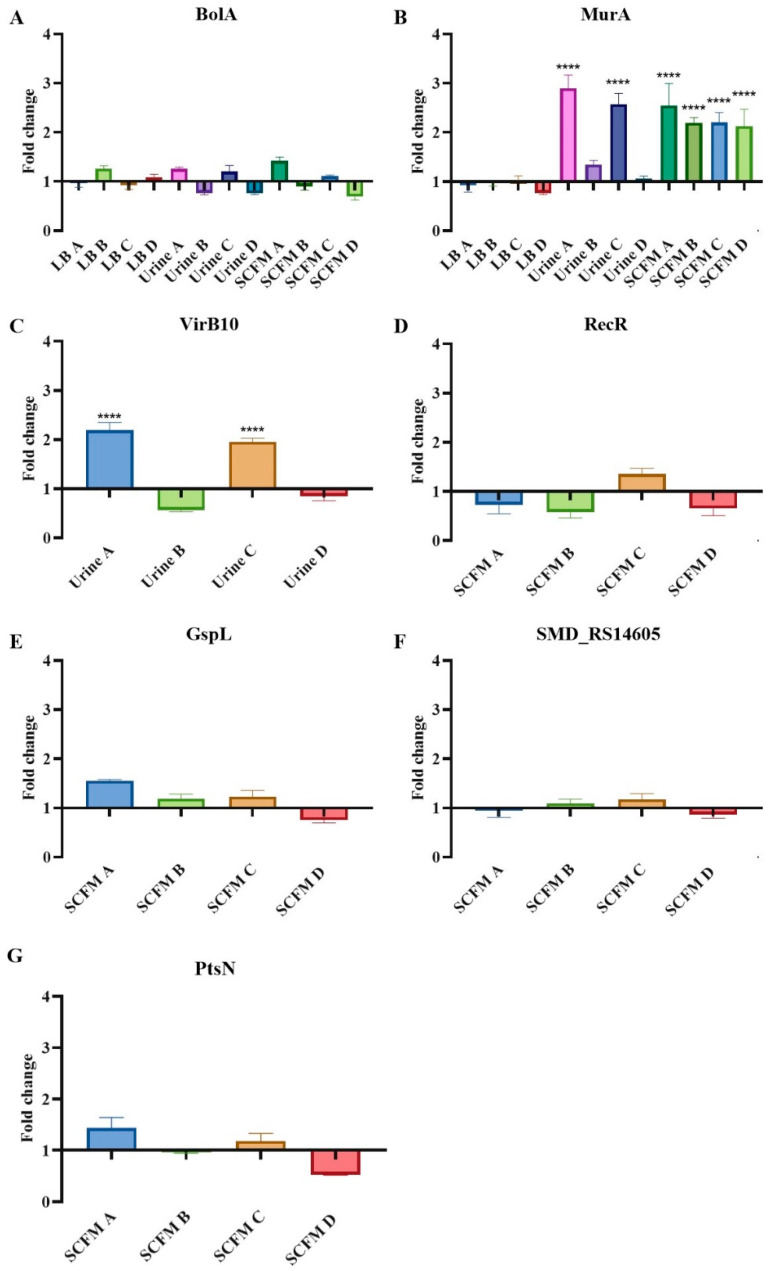
Effect of main mutations in gene expression. Expression level of genes encoding mutated genes. Fold changes in fosfomycin-evolved population were estimated regarding the expression of the D457 wild-type strain by qRT-PCR. (**A**) *bolA* expression; (**B**) *murA* expression; (**C**) *virB10* expression; (**D**) *recR* expression; (**E**) *gspL* expression; (**F**) SMD_RS14605 expression; (**G**) *ptsN* expression. Error bars indicate standard deviations of the results from three biological replicates. Statistically significant differences regarding D457 were calculated with t-test for paired samples assuming equal variances: **** *p* < 0.0001.

**Figure 4 ijms-23-01132-f004:**
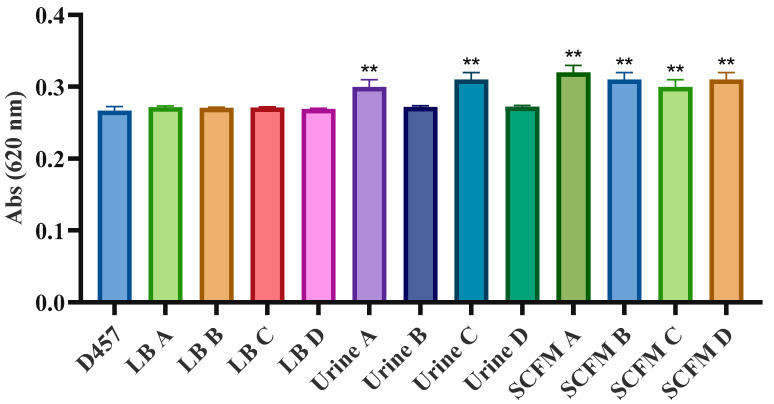
In vitro MurA activity measurements. Whole-cell lysates were preincubated with 10 mM UDP-GlcNAc, 2 mM DTT, and 50 mM Tris. To each sample, 10 mM PEP was added to start the reaction. Release of inorganic phosphate was measured by measuring the OD_620_ of the sample. Inorganic phosphate release was measured in triplicate. Statistically significant differences regarding D457 were calculated with t-test for paired samples assuming equal variances: ** *p* < 0.005.

**Figure 5 ijms-23-01132-f005:**
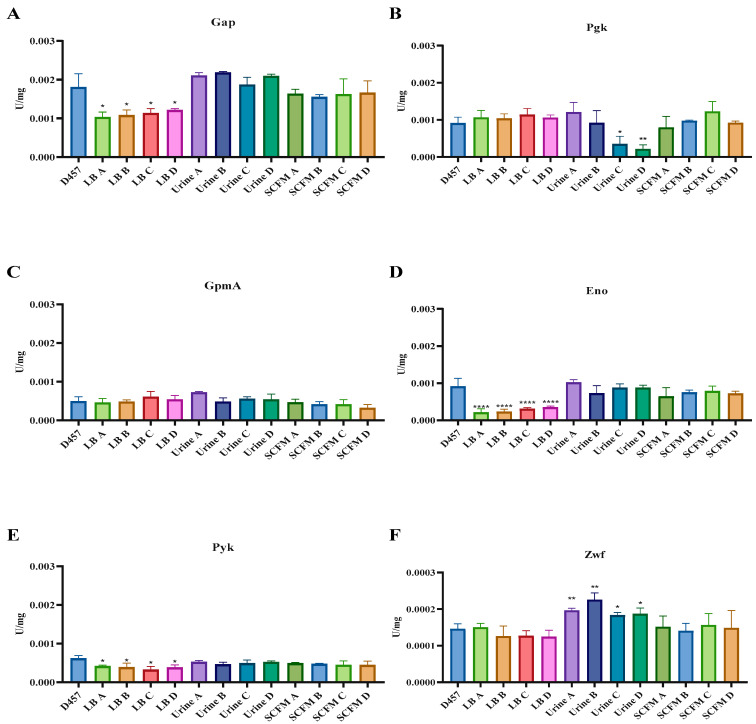
Enzymatic activity of the main enzymes of the glycolytic pathway of the D457 parental strain and the fosfomycin-evolved populations. (**A**) Zwf, glucose-6-phosphate dehydrogenase activity. (**B**) Gap, glyceraldehyde-3-phosphate dehydrogenase activity. (**C**) Pgk, phosphoglycerate kinase activity. (**D**) Gpm, phosphoglycerate mutase activity. (**E**) Eno, enolase activity. (**F**) Pyk, pyruvate kinase. Error bars indicate standard deviations for the results from three independent replicates. Values that are significantly different from the value for the wild-type D457 strain by an unpaired two-tail t-test are indicated by asterisks as follows: *, *p* < 0.05; **, *p* < 0.005; ****, *p* < 0.0001.

**Figure 6 ijms-23-01132-f006:**
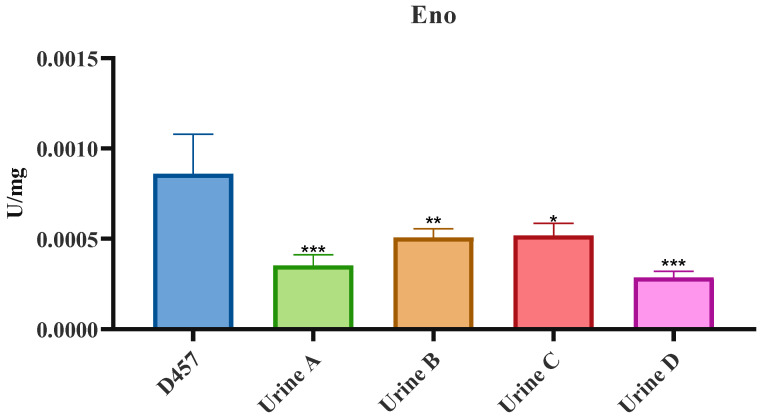
Enolase enzymatic activity of the D457 parental strain and isolated clones from the fosfomycin-evolved *S. maltophilia* populations in urine. Error bars indicate standard deviations for the results from three independent replicates. Values that are significantly different from the value for the wild-type D457 strain by an unpaired two-tail t-test are indicated by asterisks as follows: *, *p* < 0.05; **, *p* < 0.005; ***, *p* < 0.0005.

**Figure 7 ijms-23-01132-f007:**
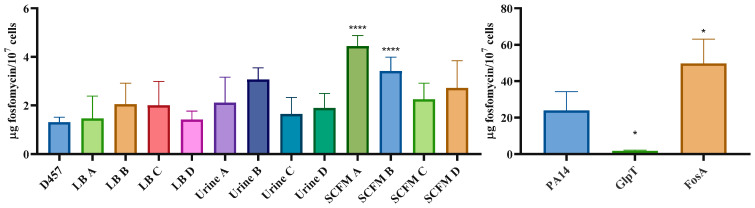
Fosfomycin-evolved populations’ intracellular concentration of fosfomycin. Comparison of the fosfomycin intracellular concentration between the fosfomycin-evolved populations and the parental strain. There is not a deficiency in the fosfomycin transport in the populations that determines its resistance, or a fosfomycin-modifying enzyme involved in this resistance. PA14 and its mutants GlpT and FosA were used as controls for the assay. The amount of intracellular fosfomycin was lower in *P. aeruginosa* when its fosfomycin transporter GlpT was inactivated. An increased fosfomycin concentration was observed in the FosA mutant relative to the parental PA14 strain. Error bars indicate standard deviations for the results from three independent replicates. Statistical significance was calculated by unpaired two-tail t-test: *, *p* < 0.05; ****, *p* < 0.0001.

**Table 1 ijms-23-01132-t001:** Fosfomycin MICs of the *S. maltophilia* evolved populations. Fosfomycin susceptibility was determined by double dilution in populations after 72 h of experimental evolution in the presence of increasing concentrations of fosfomycin.

Population	MIC (µg/mL)			
	MH	LB	Urine	SCFM
D457	256	128	256	512
LB control	256	128	128	256
LB A	16,384	>16,384	6250	>16,384
LB B	16,384	>16,384	6250	>16,384
LB C	16,384	8192	3125	>16,384
LB D	16,384	5892	3125	>16,384
Urine control	256	128	128	256
Urine A	>16,384	>16,384	>16,384	16,384
Urine B	>16,384	>16,384	>16,384	>16,384
Urine C	>16,384	>16,384	>16,384	>16,384
Urine D	>16,384	>16,384	>16,384	>16,384
SCFM control	256	128	128	256
SCFM A	>16,384	8192	6250	>16,384
SCFM B	8192	1024	3125	8192
SCFM C	8192	8192	6250	>16,384
SCFM D	16,384	1024	1562	4092

**Table 2 ijms-23-01132-t002:** WGS-identified mutations in the fosfomycin-evolved lineages.

Medium	L	Gene	Product	Localization	Type	Nucleotide Change	Amino Acid Change	Frequency (%)	Domain	Provean Score
LB	A	*eno*	Phosphopyruvate hydratase	1,828,585	SNV	T → C	Leu106Pro	93	N-terminal	−6.9
	B	*eno*	Phosphopyruvate hydratase	1,828,585	SNV	T → C	Leu106Pro	98	N-terminal	−6.9
		*phaR*	Polyhydroxyalkanoate synthesis repressor	3,219,959	SNV	C → T	Ser85Phe	22	PHB accumulation regulatory	−3.967
	C	*eno*	Phosphopyruvate hydratase	1,828,585	SNV	T → C	Leu106Pro	98	N-terminal	−6.9
	D	*eno*	Phosphopyruvate hydratase	1,828,585	SNV	T → C	Leu106Pro	94	N-terminal	−6.9
Urine	A	*bolA*	BolA family transcriptional regulator	1,158,049	SNV	C → T	Ala52Ala	28	BolA-like	0.0
		*eno*	Phosphopyruvate hydratase	1,828,585	SNV	T → C	Leu106Pro	11	N-terminal	−6.9
		*virB10*	TrbI/VirB10 family protein	2,939,074	Ins	- → C	Asp173fs	ND	Bacterial conjugation TrbI	
	B	*eno*	Phosphopyruvate hydratase	1,828,585	SNV	T → C	Leu106Pro	21	N-terminal	−6.9
		*virB10*	TrbI/VirB10 family protein	2,939,074	Ins	- → C	Asp173fs	ND	Bacterial conjugation TrbI	
	C	*bolA*	BolA family transcriptional regulator	1,158,049	SNV	C → T	Ala52Ala	12	BolA-like	0.0
		*eno*	Phosphopyruvate hydratase	1,828,585	SNV	T → C	Leu106Pro	9	N-terminal	−6.9
		*virB10*	TrbI/VirB10 family protein	2,939,074	Ins	- → C	Asp173fs	ND	Bacterial conjugation TrbI	
	D	*eno*	Phosphopyruvate hydratase	1,828,585	SNV	T → C	Leu106Pro	13	N-terminal	−6.9
		*virB10*	TrbI/VirB10 family protein	2,939,074	Ins	- → C	Asp173fs	ND	Bacterial conjugation TrbI	
SCFM	A	*gspL*	General secretion pathway protein	674,920	SNV	G → T	Val247Phe	77	Fimbrial assembly pilN	−4.162
		*ptsN*	PTS sugar transporter subunit IIA	1,152,021		AGGGCCT → GCAGGCC	GlnAlaLeu126ArgProAla	25	PTS_EIIA_2	L128A −4.542
		*bolA*	BolA family transcriptional regulator	1,158,049	SNV	C → T	Ala52Ala	36	BolA-like	0.0
		*SMD_RS14605*	ABC transporter ATP-binding protein	3,119,746	SNV	A → C	Thr419Pro	19	ABC transporter	−6.0
		*recR*	Recombination protein RecR	1,087,048	Ins	CAAGCGGGTGCCACAGAA	Upstream gene variant	ND	-	
	B	*gspL*	General secretion pathway protein	674,920	SNV	G → T	Val247Phe	64	Fimbrial assembly pilN	−4.162
		*bolA*	BolA family transcriptional regulator	1,158,049	SNV	C → T	Ala52Ala	91	BolA-like	0.0
		*recR*	Recombination protein RecR	1,087,048	Ins	CAAGCGGGTGCCACAGAA	Upstream gene variant	ND	-	
	C	*bolA*	BolA family transcriptional regulator	1,158,049	SNV	C → T	Ala52Ala	53	BolA-like	0.0
		*recR*	Recombination protein RecR	1,087,048	Ins	CAAGCGGGTGCCACAGAA	Upstream gene variant	ND	-	
	D	*bolA*	BolA family transcriptional regulator	1,158,049	SNV	C → T	Ala52Ala	87	BolA-like	0.0
		*recR*	Recombination protein RecR	1,087,048	Ins	CAAGCGGGTGCCACAGAA	Upstream gene variant	ND	-	

L: lineage; SNV: single-nucleotide variant; Ins: insertion; fs: frame shift; frequency (%): percentage of reads that contain the variation within a heterogeneous population; ND: non-determined (IGV 2.11.9 Genomics Software does not determine the frequency of insertion/deletions). PROVEN score: default threshold is −2.5, that is, variants with a score equal to or below −2.5 are considered “deleterious”, and variants with a score above −2.5 are considered “neutral”.

## Data Availability

All data used in the work have been included in the manuscript.

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
