# Peer review of "Fosfomycin Resistance Evolutionary Pathways of Stenotrophomonas maltophilia in Different Growing Conditions"

_ijms, 2022, doi:10.3390/ijms23031132_

Round 1
Reviewer 1 Report
This article provides relevant information on the mechanisms involved in the resistance to fosfomycin in Stenotrophomonas maltophilia (taking into account that they are different from those found in other gram-negative bacteria, especially in enterobacteria) and the ability of this bacterium to acquire high levels of resistance in different conditions and culture media, especially those that more faithfully represent the conditions of a true infection.
It is a very interesting work, well described and with a methodology adjusted to the demonstration of the proposed objectives. All the necessary references have been cited.
However, the appearance, in the Results section, of numerous bibliographic citations and statements that are more typical of a Discussion section, leads me to make the suggestion that both sections be combined in a single Results and Discussion section before publication.
Other minor suggestions:
- Line 167: "S. maltophilia" should be italicized.
- Section 4.1.: What bacterial inoculum was used for antibiotic susceptibility testing?
- Line 540: Please correct "are under de curve".
- Line 551: Is "from 10 g of RNA" correct?
Author Response
This article provides relevant information on the mechanisms involved in the resistance to fosfomycin in Stenotrophomonas maltophilia (taking into account that they are different from those found in other gram-negative bacteria, especially in enterobacteria) and the ability of this bacterium to acquire high levels of resistance in different conditions and culture media, especially those that more faithfully represent the conditions of a true infection.
It is a very interesting work, well described and with a methodology adjusted to the demonstration of the proposed objectives. All the necessary references have been cited.
Answer: We appreciate the positive opinion of the referee concerning our work
However, the appearance, in the Results section, of numerous bibliographic citations and statements that are more typical of a Discussion section, leads me to make the suggestion that both sections be combined in a single Results and Discussion section before publication.
Answer: We understand the point of the referee and indeed, we have combined both sections in other publications. However, we think that the Discussion section of the current article sumarises quite well the results and prefer keeping the organization of the work the way it is now-
Other minor suggestions:
- Line 167: "S. maltophilia" should be italicized.
Answer: Done
- Section 4.1.: What bacterial inoculum was used for antibiotic susceptibility testing?
Answer: Information now included
- Line 540: Please correct "are under de curve".
Answer: Done
- Line 551: Is "from 10 g of RNA" correct?
Answer: The right value is 10 µg. Thanks for the correction
Reviewer 2 Report
Dear Editor,
The article submitted by Gil-Gil T. et al., entitled "Fosfomycin resistance evolutionary pathways of Stenotrophomonas maltophilia in different growing conditions," is interesting.
The authors continued their research between Stenotrophomonas maltophilia and Fosfomycin. Actually, the use of fosfomycin for this pathogen is rare due to the low presence of S. maltophilia in the urine. A better understanding of this uncommon bacterium's acquired resistance would help everyone obtain good results in treating infections.
- Please review the affiliations
- l. 38-39 reference 7 is not adequate. Fosfomycin is one of the first-line treatments in uncomplicated UTIs - European Association of Urology Guidelines on Urinary Tract Infection - 2021.
- The introduction section should present data regarding the only use of fosfomycin in urinary tract infections and the rarity of S. maltophilia in UTI.
- l. 45 Please include the correct citation - the article has been published
- Please include Conclusions or include at the end of the Discussion section a concise message
- Please include a list of abbreviations in alphabetical order.
Author Response
The article submitted by Gil-Gil T. et al., entitled "Fosfomycin resistance evolutionary pathways of Stenotrophomonas maltophilia in different growing conditions," is interesting.
The authors continued their research between Stenotrophomonas maltophilia and Fosfomycin. Actually, the use of fosfomycin for this pathogen is rare due to the low presence of S. maltophiliain the urine. A better understanding of this uncommon bacterium's acquired resistance would help everyone obtain good results in treating infections.
Answer: We appreciate the positive opinion of the referee concerning our article
- Please review the affiliations
Answer: Done
- l. 38-39 reference 7 is not adequate. Fosfomycin is one of the first-line treatments in uncomplicated UTIs - European Association of Urology Guidelines on Urinary Tract Infection - 2021.
Answer: The referee is right, novel references, including the one suggested by the referee.
- The introduction section should present data regarding the only use of fosfomycin in urinary tract infections and the rarity of S. maltophilia in UTI.
Answer: Done
- l. 45 Please include the correct citation - the article has been published
Answer: Done
- Please include Conclusions or include at the end of the Discussion section a concise message
Answer: The last paragraph of the Discussion section summarizes the main aspects of the work
- Please include a list of abbreviations in alphabetical order.
Answer: All the abbreviations are spelled out and we do not think that such list is needed. However, if the editor believes it is worth including it, the list will be:
Amikacin (AMI)
Aztreonam (AZT)
Ceftazidime (CEFTA)
Chloramphenicol (CHL)
Ciprofloxacin (CIPRO)
Colistin (COL)
Dithiothreitol (DTT)
Embden-Meyerhof-Parnas (EMP)
General secretion pathway protein (GspL)
Gentamicin (GENTA)
Glucose-6-phosphate dehydrogenase (Zwf)
Glucose-6-phosphate transporter (UhpT)
Glutathione transferase (FosA)
Glyceraldehyde-3-phosphate dehydrogenase (Gap)
Glycerol-3-phosphate transporter (GlpT)
Horizontal gene transfer (HGT)
Insertions/deletions (INDEL)
Lipopolysaccharides (LPS)
Lysogeny broth (LB)
Minimum inhibitory concentration (MIC)
Mueller–Hinton (MH)
Nalidixic acid (NAL)
Nicotinamide adenine dinucleotide (NAD)
Nicotinamide adenine dinucleotide phosphate (NADP)
Ofloxacin (OFX)
Polyhydroxyalkanoate synthesis repressor (PhaR)
Phosphoglycerate kinase (Pgk)
Phosphoglycerate mutase (Gpm)
Phosphopyruvate hydratase (Eno)
PEP-dependent carbohydrate phosphotransferase system (PTS)
Polymerase chain reaction (PCR)
Polymyxin B (POL)
Pyruvate kinase (Pyk)
Quantitative reverse transcription PCR (qRT-PCR)
Single nucleotide polymorphism (SNP)
Streptomycin (STR)
Synthetic cystic fibrosis sputum medium (SCFM)
Tetracycline (TETRA)
Tigecycline (TIGE)
Trimethoprim/sulfamethoxazole (SXT)
Type II secretion system (T2SS)
Type IV secretion system (T4SS)
UDP-N-acetylglucosamine enolpyruvyl transferase (MurA)
Urinary tract infection (UTI)
Whole-genome sequencing (WGS)
Reviewer 3 Report
The paper is valuable and brings new information about the mechanisms underlying antibiotic resistance. I recommend extending the studies to other antibiotics as well.
Author Response
We appreciate the positive opinion of the referee concerning our work and, no doubt, the studies will be extended to other antibiotics
Reviewer 4 Report
In this publication, Gil-Gil and Martínez are addressing a highly contemporary topic, and study the patterns of antibiotic resistance from an evolutionary point of view. In this sense, the scientific approach is interesting.
The study is well designed and executed, and provides a wast arry of data that may shed more light on the mechanisms of bacterial rsistance towards antibiotics.
Nevertheless, I have several questions addressing of which would require a small revision of the manuscript:
- the authors state that of of the "growth media" was urine form 6 healthy people. Was an Ethics committee statement or informed consent not necessary for the collection? Whta was the age of the donors? Were there any inclusion/exclusion criteria for the donors?
- where/who was the wild type S. maltophilia D457 obtained from?
- what concentrations of antibiotics were used in the antibiotic susceptibility assays?
- a description of the statistical analysis is missing
- the authors could provide a short paragraph addressing the limitations of the study
- it would also be interesting to shortly discuss about the practical (clinical) outcomes and/or consequences of the data presented in the study
Author Response
In this publication, Gil-Gil and Martínez are addressing a highly contemporary topic, and study the patterns of antibiotic resistance from an evolutionary point of view. In this sense, the scientific approach is interesting.
The study is well designed and executed, and provides a wast arry of data that may shed more light on the mechanisms of bacterial rsistance towards antibiotics.
Answer: We appreciate the positive opinion of the referee concerning our work
Nevertheless, I have several questions addressing of which would require a small revision of the manuscript:
- the authors state that of of the "growth media" was urine form 6 healthy people. Was an Ethics committee statement or informed consent not necessary for the collection? Whta was the age of the donors? Were there any inclusion/exclusion criteria for the donors?
Answer: Donors were 6 healthy people (3 men and 3 women) that were between 25 and 55 years old. The exclusion criterium was prior antibiotic treatment: none of the donors had received an antibiotic treatment in the last 6 months. These features are now stated. Informed consent was obtained.
- where/who was the wild type S. maltophilia D457 obtained from?
Answer: The strain D457 is a clinical isolate obtained from a bronchial aspirate that has been used as model strain in several publications. This information as well as a reference describing the strain in more detail is now included.
- what concentrations of antibiotics were used in the antibiotic susceptibility assays?
Answer: MIC test strips (Liofilchem) where used for all the susceptibility assays except from vancomycin susceptibility assay in which concentrations from 4092 to 4 ug/mL were used.
- a description of the statistical analysis is missing
Answer: The statistical analysis was based in t-test for paired samples assuming equal variances. This feature is stated in the legends of all figures where statistical significance is discussed.
- the authors could provide a short paragraph addressing the limitations of the study
Answer: A short paragraph indicating that S. maltophilia rarely produce UTIs, a feature that might be a limitation of the current study has been included
- it would also be interesting to shortly discuss about the practical (clinical) outcomes and/or consequences of the data presented in the study
Answer: The last paragraph discusses the practical consequences of the study, highlighting that evolution towards AR may differ depending on the infection point.